# Learning Objectives Matrix in DIM.RUHR: A Didactic Concept for the Interprofessional Teaching of Data Literacy in Outpatient Health Care

**DOI:** 10.3390/healthcare13060662

**Published:** 2025-03-18

**Authors:** Vivian Lüdorf, Anne Mainz, Sven Meister, Jan P. Ehlers, Julia Nitsche

**Affiliations:** 1Didactics and Educational Research in Health Care, Faculty of Health, Witten/Herdecke University, 58448 Witten, Germany; jan.ehlers@uni-wh.de (J.P.E.); julia.nitsche@uni-wh.de (J.N.); 2Health Informatics, Faculty of Health, Witten/Herdecke University, 58448 Witten, Germany; anne.mainz@uni-wh.de (A.M.); sven.meister@uni-wh.de (S.M.); 3Department of Healthcare, Fraunhofer Institute for Software and Systems Engineering ISST, 44147 Dortmund, Germany

**Keywords:** healthcare data management, data literacy, outpatient healthcare, digital transformation, didactic concept, learning objectives matrix, interprofessional teaching

## Abstract

(1) **Background**: Each year, significant volumes of healthcare data are generated through both research and care. Since fundamental digital processes cannot function effectively without essential data competencies, the challenge lies in enhancing the quality of data management by establishing data literacy among professionals in outpatient healthcare and research. (2) **Methods**: Within the DIM.RUHR project (Data Competence Center for Interprofessional Use of Health Data in the Ruhr Metropolis), a didactic concept for interprofessional data literacy education is developed, structured as a learning objectives matrix. Initially conceived through a literature review, this concept has been continually developed through collaboration with interprofessional project partners. The study was conducted between February 2023 and June 2024. (3) **Results**: The foundational structure and content of the didactic concept are based on various scientific studies related to general data literacy and the outcomes of an interactive workshop with project partners. Eight distinct subject areas have been developed to encompass the data literacy required in healthcare professions: (1) Fundamentals and general concepts, (2) ethical, legal, and social considerations, (3) establishing a data culture, (4) acquiring data, (5) managing data, (6) analyzing data, (7) interpreting data, and (8) deriving actions. Within these, learners’ data literacy is assessed across the four competency areas: basic, intermediate, advanced, and highly specialized. (4) **Conclusions**: The learning objectives matrix is anticipated to serve as a solid foundation for the development of teaching and learning modules aimed at enhancing data literacy across healthcare professions, enabling them to effectively manage data processes while addressing the challenges associated with digital transformation.

## 1. Introduction

Proficiency in managing digital technologies has become crucial in healthcare to effectively drive digital transformation, especially as the sector transitions from traditional analog systems to advanced digital technologies. The extensive adoption of digital devices and big data management systems, such as hospital information systems and practice management systems, has fundamentally reshaped how patient information is collected, stored, and utilized [1,2]. The rapid pace of technological advancement underscores the necessity for healthcare professionals to continuously enhance their digital skills to keep up with technological advancements [3]. As innovations, like telemedicine platforms and integrated digital patient records, become increasingly prevalent, the ability to effectively engage with these technologies is essential for enhancing the quality of care, facilitating accurate and timely diagnoses, and tailoring treatment plans to individual patient needs [4].

Each year, vast volumes of patient-related data are generated. The effectiveness of systems like electronic health records, which consolidate diverse health data into a single, accessible format, relies on healthcare providers’ ability to interpret and use this data efficiently [5]. As data becomes increasingly central to healthcare delivery, sufficient data literacy is crucial for addressing the increasing demands of data management [6]. Ridsdale et al. [7] define general data literacy as the ability to critically collect, manage, evaluate, and apply data. This comprehensive skill set enables individuals to integrate data into everyday thinking and decision-making processes, addressing real-world problems [8]. As interactions with data become more frequent, data literacy is now recognized as a vital life skill, with individuals more often making decisions based on data and managing their personal data [8]. This competency involves various skills, including the identification, collection, organization, analysis, summarization, and prioritization of data [9].

Despite its increasing significance, there is no universal definition of data literacy in healthcare settings. In this context, the term “data literacy” is preferred over broader terms such as “competence” or “competencies” to emphasize the specific skill set required for effectively managing and utilizing data in healthcare contexts. While “competence” can refer to general abilities across various domains, “data literacy” captures the critical and specialized skills necessary for interpreting, analyzing, and applying data effectively [7,8], particularly in relation to patient information and healthcare delivery. This distinction ensures clarity in addressing the unique challenges and requirements associated with data management in the medical field.

In addition, related terms such as big data skills, eHealth literacy, and digital literacy—encompassing not only the use of digital technologies but also the knowledge and skills needed to understand, interpret, and securely manage data—are often used interchangeably to data literacy without a consistent framework or clear understanding of how they are interconnected [10,11]. This variation in terminology reflects the broad and evolving nature of the concept, covering a range of skills and knowledge necessary for the effective use of data.

In medical contexts, data literacy empowers healthcare providers to navigate complex data environments, ensuring that the insights derived from patient data are accurate and actionable [12]. Moreover, as these systems often involve sensitive patient information, data literacy is essential for maintaining compliance with stringent data privacy regulations and ensuring the security and confidentiality of patient records [13]. In outpatient healthcare and research, proficient data literacy is crucial for several reasons. First, healthcare professionals are required to manage and utilize extensive patient-related data, which demands not only technical skills but also a thorough understanding of data ethics and contextual application [14]. Second, research involving healthcare data often relies on high-quality data for accurate findings and reliable conclusions [15]. Therefore, any discrepancies or deficiencies in data handling among healthcare professionals can significantly reduce the quality of data management, ultimately compromising patient care and research outcomes.

Although data literacy, therefore, can be considered one of the foremost key competencies of the future in the healthcare sector, it is currently neither embedded in medical school curricula nor in the training regulations of various health professions in Germany [16]. As a result, there are only a few initiatives aimed at comprehensively implementing data literacy in outpatient healthcare and research. This gap results in varying levels of data literacy among healthcare professionals, impacting the consistency and quality of data handling practices. Additionally, the rapid pace of digital transformation in healthcare means that existing training programs may quickly become outdated, necessitating ongoing updates to educational content and methods [17].

Given the importance of fundamental data literacy in the medical sector, this issue raises the question of how to systematically establish comprehensive data literacy across various professions in outpatient healthcare and research. 

The project DIM.RUHR (Data Competence Center for Interprofessional Use of Health Data in the Ruhr Metropolis) (grant number: 16DKZ2008A), funded by the German Federal Ministry of Education and Research (BMBF), financed by the European Union—NextGenerationEU and launched in 2023, demonstrates how to address these challenges and establish a solid foundation for developing data literacy in the medical sector. The Data Competence Center places a strong emphasis on fostering data literacy as a key goal of the project, aiming to equip professionals in outpatient healthcare and research with the necessary skills to handle health data effectively. Central to this initiative is the development of interprofessional learning and teaching modules that will enhance the data literacy of healthcare professionals across diverse fields such as medicine, social sciences, informatics, and public health.

The project recognizes that disparities in data literacy currently affect the quality of health data management, including data generation, storage, processing, and use. By addressing these gaps, the initiative aims to standardize practices and clarify responsibilities for data management. To achieve this, DIM.RUHR will design a comprehensive didactic framework that supports the creation and implementation of teaching and learning modules tailored to the specific needs of various healthcare professions. This framework represents an initial theoretical construct, with its evaluation planned as part of later project stages. Teaching and learning modules will be piloted to ensure the framework’s practical applicability and to incorporate feedback for continuous improvement. Consequently, empirical testing of the framework is yet to be carried out.

This educational concept is critical for building long-term competence in data management, ensuring that professionals from different sectors can collaborate effectively in managing outpatient healthcare data, contributing to improved research, patient care, and health outcomes.

Considering these aspects, the following research question can be derived: how must a didactic concept be designed to serve as a foundation for the further development of teaching and learning modules and establish comprehensive data literacy across the professions in outpatient healthcare and research?

## 2. Materials and Methods

The present approach follows the research design of intra-method triangulation, which involves the combination of various methodological approaches within qualitative research [18]. In this context, extensive literature research on general data literacy, combined with the outcomes of an interactive workshop with interprofessional project partners as well as conducting expert interviews with several healthcare and research professionals, led to the development of the core framework for the didactic concept, which is represented by a learning objectives matrix (LOM). The study was carried out from February 2023 to June 2024. The didactic concept serves as a structured framework for systematically integrating data literacy into outpatient healthcare and research education. It outlines specific learning objectives, competency levels, and instructional methods to ensure that healthcare professionals acquire the necessary skills for managing health data effectively. By utilizing the LOM, the concept provides a targeted approach to teaching and assessing data literacy competencies, aligning them with the real-world requirements of different professional groups. This framework enables a standardized and adaptable method for embedding data literacy into professional training and education programs. The individual steps are outlined in the following sections.

To conceptualize the LOM, the following methodological steps were initially required: Before constructing this didactic framework, two thorough literature reviews were conducted to assess the current state of research on general data literacy. The first literature review was carried out in February 2023, focusing on the terms “data literacy” and “framework/mode” to produce a working definition that outlines the essential skills for effective data management. To ensure a broad and open search for relevant literature using the selected keywords, Google Scholar was utilized as the primary database for conducting the literature review. Publications prior to 2013, those in languages other than German or English, and those specifically targeting teachers and librarians were excluded. The exclusion of publications prior to 2013 was based on the recognition that the concept of “data literacy” has evolved significantly, especially in the last decade. Earlier sources often lack alignment with contemporary understandings of data literacy, which are shaped by the rapid technological and digital transformation and the emergence of data-driven frameworks. By focusing on literature from 2013 onward, the review ensures relevance to the current discourse and avoids outdated perspectives that may not reflect the skills and challenges associated with digitalization and modern guidelines of data protection. To expand on the initial review, a second literature review conducted from January to February 2024 broadened the findings by incorporating additional terms such as “digital literacy in healthcare”, “eHealth”, and “big data (skills) in healthcare”, using the same exclusion criteria. The literature identified in this process provides a structural template for the LOM and offers content-related guidance.

The working definition of data literacy used in this study was derived from the extensive literature reviews and is based on a synthesis of the definitions provided by Ridsdale et al. [7], Grillenberger and Romeike [19], and Schüller [20]. According to these authors, data literacy encompasses the knowledge, skills, and attitudes/values necessary for the effective planning, execution, and improvement of all process steps involved in deriving valuable insights or making decisions from data. This takes place within a collaborative culture that adheres to ethical, legal, and social considerations. This working definition of data literacy was integrated into the LOM in such a way that all subsequently formulated learning objectives are based on this definition. The fundamental structure of the LOM was adapted from the LOM on research data management by Petersen et al. [21] and specifically tailored to the requirements of health data utilization. To create a comprehensive and practical foundation, the matrix was enriched with learning objectives derived from the content of scientific literature on general data literacy education. These contributions include works by Engelhardt et al. [22], Heidrich et al. [23], Ridsdale et al. [7], Schüller [20], and Vuorikari et al. [24].

Another central component of the methodology involves a digital workshop, which was conducted as an internal project discussion within the DIM.RUHR consortium in February 2023. This workshop did not follow a focus group format but served as a structured exchange among consortium members from various disciplines. Workshop participants engaged in group discussions to identify the essential knowledge, skills, and values required for developing robust data literacy. The invitations to participate were extended via email to all consortium members based on their expertise in relevant fields. The results from the digital workshop were documented using a shared Miro board, where all participants could collaboratively add content. To ensure clarity and structured discussions, the workshop was divided into eight breakout sessions. Rather than conducting a traditional content analysis, the comments and inputs recorded on the Miro board were directly utilized to refine the LOM by adding or modifying content as needed in later stages.

In the further conceptualization process of the LOM, various subject areas were developed to reflect and comprehensively address the specific requirements for the utilization of health data. These areas include an evaluation system to capture the varying levels of data literacy among learners. In the context of the revision, project members decided against categorizing the various learning levels by educational degrees. Instead, the approach now involves stratifying the LOM by competency levels, which enables the classification of participants in the data utilization process based on their individual tasks, skills, and prior knowledge.

To tailor the matrix to specific professions in outpatient healthcare and research, five target group-oriented personas were created in April 2024 to help identify and integrate relevant data literacy into the document: General practitioner, medical student, student of medical professions (e.g., healthcare, nursing science, midwifery science), medical assistant, and young researcher. These personas represent typical professional groups and their specific data-related requirements in healthcare. Within the revision process, all data competencies identified as crucial for the personas were integrated into the LOM and thoroughly reviewed for completeness. In a recurring and continuous process of alignment, the requirements faced by professionals in the healthcare fields addressed by the project were examined, and the skills needed to adequately manage these challenges were identified.

A qualitative approach was chosen for the development of well-founded personas. To achieve this, targeted face-to-face interviews were conducted with professionals from relevant healthcare occupations in May 2024. Participants were recruited through the professional networks of the interviewers, including teaching activities, colleagues, and relevant online forums. The selection criteria ensured that participants represented the occupational groups relevant to the persona development. The interviews were conducted across multiple project sites by experts from DIM.RUHR. During the interviews, the participants were asked about the demands and required competencies they encounter in their professional practice. The individual statements were subsequently aligned with the previously developed personas. A classical content analysis was not performed for the interview results. Instead, the participants were shown a profile of the developed personas, and discussions were held regarding which content the various professional groups deemed necessary to complete the personas. The profile was collaboratively filled in during the interviews, with participants contributing their insights in real time. These profiles were then used in the subsequent stages of the development process to align the content with the learning objectives of the LOM. This phase provided direct feedback from practitioners, ensuring that the personas and the competencies within the LOM were grounded in real-world experience and accurately reflected the practical challenges faced by healthcare professionals.

In a final validation step, a cross-referencing process was conducted to ensure that the developed LOM accurately represents a diverse range of competencies relevant to different medical fields. To achieve this, learning objective catalogs from various healthcare professions were systematically searched and analyzed. The primary search was conducted using Google in May 2024, focusing on official training and education regulations applicable to the professional groups addressed in the LOM.

For each identified catalog, a targeted content analysis was performed to extract references to data-related competencies. This analysis specifically aimed to determine the contexts in which students or trainees are expected to engage with data, ensuring that relevant competencies were incorporated into the LOM. The integration of identified content from the learning objective catalogs into the matrix took place between May and June 2024.

In summary, the integration of the individual steps listed above into the overall structure of the LOM, as illustrated in Figure 1, proceeded as follows: Following an extensive literature review, A.M. merged the tabular framework from the literature review into a new comprehensive document, incorporating the relevant learning objectives and copying relevant passages. The table was subsequently cross-referenced with additional matrices and examined for potential duplications concerning key aspects of data literacy. Based on the workshop results, A.M., S.M., J.E., and J.N. integrated the developed thematic areas and the subdivision by competency levels into the existing document. In a subsequent step, V.L. reviewed the document and reorganized all listed learning objectives with regard to their appropriate competency levels. To complete the synthesis of the individual steps into the overall LOM, V.L. evaluated the learning objectives presented in the matrix for their congruence with the developed personas and the learning objective catalogues to ensure that they adequately reflect the requirements of the respective professions in outpatient healthcare and research. 

While Figure 1 suggests a chronological sequence, the development process was equally characterized by iteration and reciprocity. Individual stages built upon one another in a sequential manner, yet transitions between them were fluid and occurred repeatedly throughout the entire process, reflecting the dynamic nature of conceptual refinement.

## 3. Results

Given the complexity and scope of the project, the full English translation of the LOM is provided as Appendix A.

### 3.1. Structuring the LOM

During the internal project workshop, the basic structure of the LOM was established. As illustrated in Figure 2 below, the following representatives from various disciplines participated in the workshop (*N* = 34, 44.12% female): health sciences (*n* = 6), general medicine (*n* = 5), health informatics (*n* = 4), software and systems engineering (*n* = 4), didactics and educational research (*n* = 3), health services research (*n* = 3), law (*n* = 3), social sciences (*n* = 2), media education (*n* = 2), and information sciences (*n* = 2). Due to the voluntary nature of participation, socio-demographic data such as age were not fully collected, leading to major potential distortions in the average age as some participants declined to provide this information.

Initially, the workshop participants discussed the stratification of various competencies following the proficiency levels outlined in DigComp 2.2 (EU reference framework for digital skills) [24]. Consequently, the original eight levels were consolidated into four levels, which can be described as follows. (1) Basic: learners can perform fundamental tasks independently or with guidance. (2) Intermediate: learners can independently solve well-defined problems or with appropriate guidance. (3) Advanced: learners can impart knowledge to others and solve advanced problems independently in complex contexts. (4) Highly specialized: learners can operate at a highly specialized level, transfer their skills to new domains, and support others in their work and problem-solving.

In addition to the four competency levels, the workshop participants classified the respective learning objectives into the competency areas of knowledge (K), skills (S), and attitudes (A). These competency areas align with the three dimensions outlined by Schüller [21]. Knowledge and skills were categorized as cognitive learning domains, while attitudes were defined as affective learning domains. Although cognitive and affective competency areas are interdependent [25], these domains are organized separately within the LOM to enhance clarity.

The workshop participants engaged in a deliberation to identify the subjects that should be incorporated into the LOM. To ensure a comprehensive representation of the data competencies essential for various healthcare professions, it was concluded that the framework should encompass eight distinct subject areas with various subcategories, which can be found in Figure 3 below: (1) fundamentals and general concepts, (2) ethical, legal, and social considerations, (3) establishing a data culture, (4) acquiring data, (5) managing data, (6) analyzing data, (7) interpreting data, and (8) deriving actions. These subject areas categorize data literacy into the four competency levels based on the required knowledge, skills, and attitudes/values.

### 3.2. Finalization of the LOM

Through the final alignment with the developed personas and various learning objective catalogs in a subsequent editing process, the learning objectives represented in the LOM have been created to precisely meet the needs and requirements of healthcare professionals.

Based on the conducted interviews (*N* = 21), key requirements for data competencies in various healthcare professions were identified. The sample consisted of 71% female participants, with a mean age of *M* = 35.95 years. Participants included medical assistants (*n* = 4), general practitioners (*n* = 5), medical students (*n* = 4), healthcare students (*n* = 3), and health services researchers (*n* = 5). The interviews were predominantly conducted face-to-face as individual sessions, with one group interview conducted upon participants’ request. The results indicate that the specific requirements for competent data handling vary depending on the professional group. While medical assistants primarily emphasized practical documentation and administrative tasks, general practitioners focused on integrating digital patient records and evidence-based decision-making. Additionally, students from both groups expressed an increased need for training in data interpretation and security.

As a result of the current research process, a didactic concept for interprofessional data literacy education in outpatient healthcare and research has been developed, which is represented by the LOM. Through the systematic integration of these methodological steps, the development of the LOM followed an iterative approach. The literature review provided the theoretical foundation, while the workshop discussions established the structural framework. Subsequently, expert interviews were conducted to validate the identified competencies, ensuring their alignment with the real-world requirements of healthcare professionals. The insights from these interviews informed the refinement of the LOM before it was aligned with existing learning objectives catalogs. The final cross-referencing with these catalogs ensured consistency with educational standards and further strengthened the practical applicability of the framework.

The cross-referencing process led to the identification of 15 learning objective catalogs representing the training and education regulations of the targeted professional groups. These included national and institutional frameworks such as the National Competence-Based Learning Objectives Catalog for Medicine (NKLM) for medical students, study regulations for midwifery and nursing sciences, and training and examination regulations for prospective medical assistants. The analysis of these catalogs confirmed that data-related competencies are explicitly addressed in various healthcare education frameworks. The identified competencies primarily encompassed data protection, ethical and patient-centered data handling, as well as the analysis and interpretation of patient data. The integration of these competencies into the LOM ensures that the matrix aligns with existing educational frameworks and professional expectations. Additionally, the results indicate that while some professions explicitly outline data-related competencies, others lack detailed specifications in this area. This underscores the need for a standardized approach to embedding data literacy into healthcare education. The LOM thus serves as a structured framework for addressing these gaps and fostering interdisciplinary consistency in data-related training.

Through its structure and the target group-specific learning content, it provides the foundation for developing teaching and learning modules, which will be primarily published as open educational resources during the subsequent phases of the DIM.RUHR project. In the future, the LOM may also serve as the foundation for designing specialized training programs aimed at enhancing comprehensive data literacy among healthcare professionals and students, thereby equipping them to address the challenges posed by the rapidly advancing digital transformation in outpatient healthcare and research. While the complete document of the LOM can be found in Appendix A, an extract of the LOM from the first subject area, “fundamentals and general concepts”, is illustrated in Figure 4 below.

## 4. Discussion

### 4.1. Key Insights and Their Meaning

Enhancing data literacy in outpatient healthcare and research is critical for navigating the digital transformation and optimizing data management practices to achieve better patient and research outcomes. As healthcare systems transition from analog to advanced digital technologies, the ability of professionals to effectively manage and utilize large volumes of data becomes progressively more critical. Data literacy is not only crucial for understanding and analyzing health data but also for leveraging digital tools such as public health information systems and electronic health records [1]. Proficient data literacy allows professionals to utilize these technologies to improve care quality, ensure accurate and timely diagnoses, and customize treatment plans to meet individual patient needs [4].

Despite the acknowledgment of data literacy as a critical future competency in the healthcare sector, it remains largely absent from medical school curricula and the training regulations of various health professions in Germany [16]. This lack of integration has led to limited initiatives focused on comprehensively embedding data literacy in outpatient healthcare and research. As a result, there are significant disparities in data literacy among healthcare professionals, which negatively impacts the consistency and quality of data management practices. Insufficient data literacy among healthcare professionals may impede the ability to effectively utilize modern technologies, which could lead to compromised patient care quality and reduced accuracy in medical decision-making [26].

To address this issue and lay a solid foundation for the future implementation of data literacy in outpatient healthcare and research, it is essential to design a didactic concept that equips healthcare professionals with comprehensive skills in managing data effectively. This paper, therefore, investigated the critical question of how such a didactic concept should be designed to serve as a foundation for the further development of teaching and learning modules, thereby establishing comprehensive data literacy across professions in outpatient healthcare and research.

The LOM developed within the DIM.RUHR project addresses the challenge of inconsistent and often inadequate data literacy in outpatient healthcare and research, with the goal of closing this gap. Through an extensive literature review, a collaborative workshop, and ongoing cooperation among all project participants, the LOM was developed to serve as the didactic foundation for the future conceptualization of open-access teaching and learning modules. In addition, the creation of five personas and the subsequent alignment with the matrix ensured that the learning objectives across different subject areas were appropriately formulated and tailored to the specific needs of healthcare professionals. This approach contributes to the existing literature by providing a comprehensive framework for integrating data literacy into healthcare education. By combining expert interviews, cross-referencing with established learning objective catalogs, and drawing on contemporary research, the study offers a structured methodology that addresses current challenges in health data management and lays the groundwork for future educational resources.

As a result, it is expected that the LOM will provide a solid foundation for the development of teaching and learning modules designed for interprofessional data literacy training among professionals in health services research and outpatient care. By tailoring the modules to the specific needs and target groups of various professions within outpatient healthcare and research, it is anticipated that these modules can significantly enhance the quality and effectiveness of healthcare data management. The design of teaching and learning modules to strengthen data literacy in outpatient healthcare and research is a crucial step towards adequately addressing the growing demands of managing healthcare-related data in the future [27].

Given the persistent discrepancies in data handling among the various professions in outpatient healthcare and research, this further underscores the necessity of integrating data literacy into the curricula and training regulations of healthcare professions to effectively address the evolving challenges of digital transformation [3]. The curricular integration of data literacy is essential for fostering a workforce equipped to handle the growing complexities of healthcare data management [28]. The inclusion of data literacy in educational standards would ensure that healthcare professionals at all levels are comprehensively prepared to engage in data-driven decision-making, ultimately leading to improved patient outcomes and more robust research findings [16].

Training programs can substantially enhance the data literacy of students and healthcare professionals [29,30]. One example is “dipraxis”, a German program by the Kassenärztliche Vereinigung Westfalen-Lippe (KVWL), which provides healthcare professionals with practical training to enhance their digital competencies. Additionally, the KVWL offers other specialized training programs, such as the “Digi-Managerin”, which specifically prepares medical assistants to become Digi-Managers. These professionals serve as key contacts for digitalization initiatives, ensuring the integration of data-driven processes and providing targeted training tailored to the needs of outpatient healthcare settings. Both programs emphasize the importance of a practical, role-specific approach in advancing data literacy and fostering digital transformation within healthcare. By implementing targeted training programs and conducting ongoing research, outpatient healthcare professionals and researchers can better meet the demands of data-driven decision-making and contribute to the advancement of patient care and scientific knowledge [31]. 

The LOM developed within DIM.RUHR could serve as a foundational step in curricular integration and design of training programs, providing a structured approach for conceptualizing targeted teaching and learning modules. The learning objectives are designed to address the specific data-related competencies required by various professions in outpatient healthcare and research, ensuring that training is both relevant and practical. By embedding data literacy into the formal education and training of healthcare professionals, the project aims to create a sustainable framework for ongoing data literacy development. This approach could not only prepare new graduates for the demands of modern healthcare environments but also support the continuous professional development of current practitioners [32].

Additionally, the LOM could function as a solid framework for the development of assessments and certifications, thereby ensuring that the competencies acquired by healthcare professionals meet established standards. By aligning assessment methods and certification criteria with the specific learning objectives defined in the matrix, educators can establish a cohesive evaluation system that offers a standardized measure of data literacy levels across various healthcare professions. The matrix can precisely outline the necessary skills and knowledge, allowing the development of targeted assessments and certification programs.

### 4.2. Limitations

Although the LOM provides a robust and target-audience-appropriate foundation for the future development of teaching and learning modules, with its objectives formulated through a comprehensive process and constant reference to healthcare professions, it nonetheless exhibits some limitations. Firstly, the matrix represents a theoretical construct that has yet to be tested in real-world healthcare contexts. Consequently, empirical evaluation of the framework is yet to be carried out. Furthermore, the constrained findings of the literature review can also be identified as a limitation, as there is currently only a small number of frameworks available to assess data literacy. At present, no studies on the acquisition of data literacy in outpatient healthcare settings exist. Additionally, the five personas developed do not encompass all professions within outpatient healthcare and research, potentially limiting the practical applicability of the LOM. Besides that, the learning objectives outlined in the matrix primarily focus on research data, which may differ from the health data encountered in everyday healthcare practice. Given the current lack of a unified definition of data literacy and its overlap with terms such as digital literacy, big data skills, and eHealth literacy, achieving a comprehensive focus on data processes in literature remains challenging, as these terms also include, e.g., the management of technological innovations and healthcare engagement.

### 4.3. Outlook and Implications

To validate and evaluate the theoretical framework of the LOM (the detailed planning for validation and evaluation is still in progress), teaching and learning modules will be developed on its basis in the further course of the project. These will be published as open educational resources in the form of learning modules. The goal is to make these modules freely accessible to all healthcare professionals, thereby promoting and establishing interprofessional data literacy throughout the entire healthcare sector. Moreover, it is planned to present the entire LOM on the project homepage. A search function will be developed to enable filtering of the necessary data competencies by professional groups within outpatient healthcare settings. This can primarily be accomplished through the prior development of the various personas and will ensure that the specific data competencies required for each profession are immediately identifiable.

In the future, continuous evaluation and research are essential for refining the educational approach the LOM represents. Future studies should examine the impact of data literacy training on healthcare outcomes and identify the most effective methods for delivering this education across various healthcare professions. By persistently adapting and enhancing the curriculum to align with the evolving demands of the healthcare sector, these efforts will help ensure that healthcare professionals are adequately prepared to navigate the data-driven future of healthcare. Additionally, research should investigate the long-term benefits of enhanced data literacy on healthcare practices and research productivity.

## 5. Conclusions

In conclusion, the development and implementation of teaching and learning modules aimed at enhancing data competencies are crucial for addressing the growing complexities of healthcare data management. In the future, by promoting interprofessional collaboration and aligning with the specific needs of healthcare professions, these modules have the potential to significantly improve the quality and effectiveness of data management in both outpatient care and research.

Enhancing data literacy in outpatient healthcare and research is essential for successfully navigating digital transformation and improving data management practices. This study highlights the current gaps in data literacy education and training, emphasizing the need for structured teaching and learning modules tailored to the specific needs of healthcare professionals. The LOM developed within the DIM.RUHR project provides a comprehensive framework for addressing these challenges and serves as a foundation for future curricular integration. By aligning data literacy training with professional requirements, the LOM has the potential to standardize competencies and improve data-driven decision-making across healthcare professions. However, further empirical validation and real-world implementation are necessary to assess its practical effectiveness. The integration of data literacy into formal education and continuous professional development will be crucial for ensuring long-term improvements in patient care and research quality. Moving forward, sustained research and targeted training initiatives will be key to fostering a data-literate healthcare workforce capable of meeting the demands of a rapidly evolving digital landscape.

## Figures and Tables

**Figure 1 healthcare-13-00662-f001:**
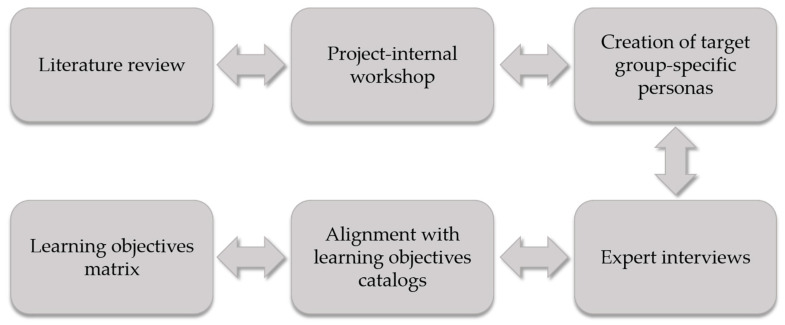
Overview of the development process of the learning objectives matrix.

**Figure 2 healthcare-13-00662-f002:**
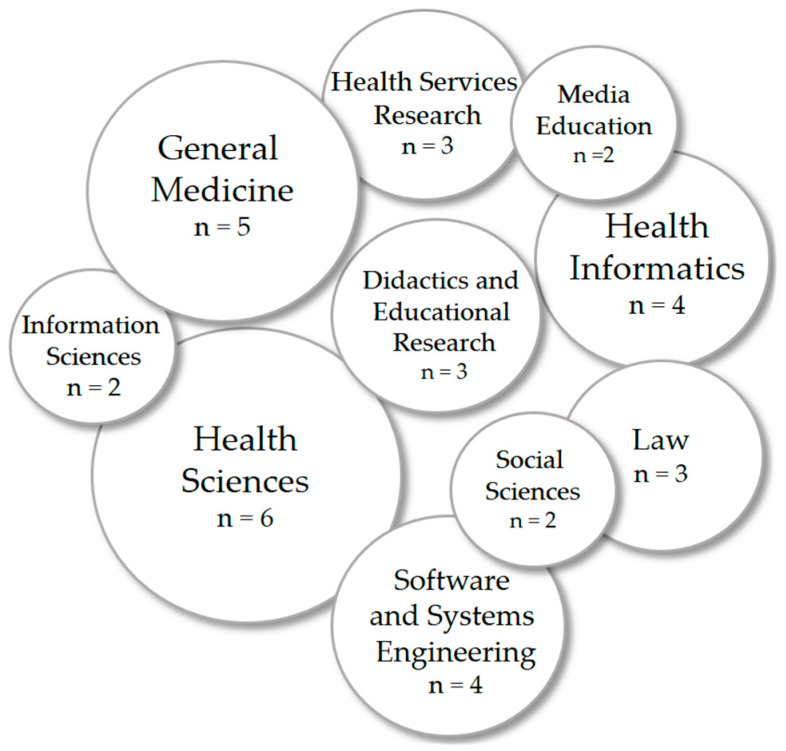
Presentation of the disciplines involved in the development of the learning objectives matrix (*N* = 34).

**Figure 3 healthcare-13-00662-f003:**
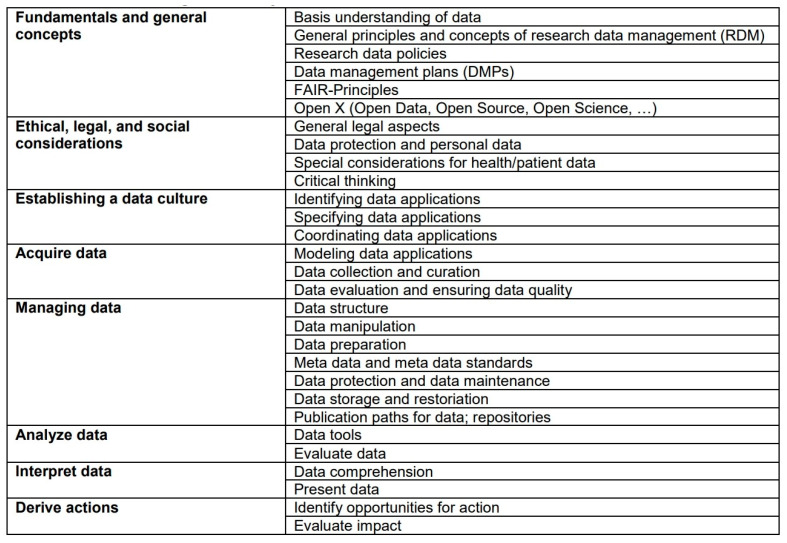
Overview of the integrated subject areas of the LOM.

**Figure 4 healthcare-13-00662-f004:**
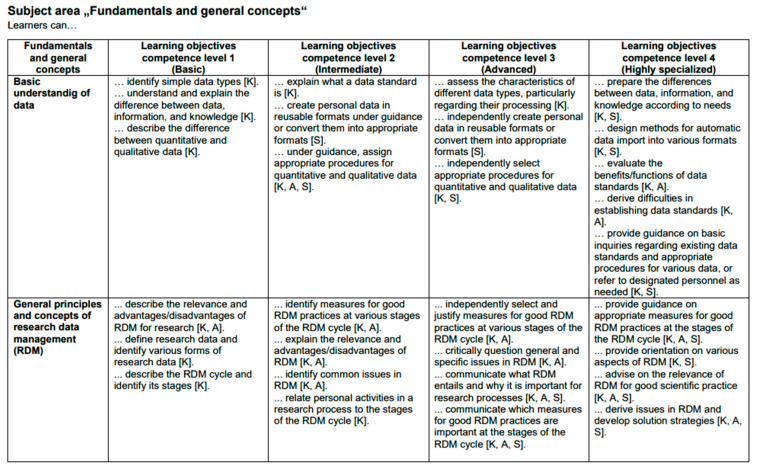
Extract of the learning objective matrix from the subject area “Fundamentals and general concepts”.

## Data Availability

The original contributions presented in the study are included in the article/Appendix A, further inquiries can be directed to the corresponding authors.

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
