# Peer review of "Learning Objectives Matrix in DIM.RUHR: A Didactic Concept for the Interprofessional Teaching of Data Literacy in Outpatient Health Care"

_healthcare, 2025, doi:10.3390/healthcare13060662_

Round 1
Reviewer 1 Report
Comments and Suggestions for Authors
- The resolution of all figures is relatively low, with some being unreadable. This is not acceptable for journal submission.
- Line 55: What is the purpose of the highlight? The same applies to the rest of the manuscript.
- The research methodology is not well established, making the reproducibility of the project questionable.
- The study is heavily qualitative in nature.
- The findings are difficult to justify in relation to the methodology. A direct mapping between the methodology and the results/discussion may be necessary.
Author Response
Thank you very much for taking the time to review this manuscript. Please find the detailed responses in the attachment.

Reviewer 2 Report
Comments and Suggestions for Authors
Overall the manuscript is an interesting paper. However, for improvement
1) Give a brief explanation about the "didactic concept'
2) The figures are unclear, therefore need to re-draw.
3) This sentence is a bit confusing:
Although the process depicted in Figure 2 appears to follow a chronological sequence, it is better understood as a reciprocal process. While individual stages build on one another, transitions between stages occurred frequently throughout the entire development process.
Comments on the Quality of English LanguageIn my opinion, the quality of the English language is Good because the writing is quite
clear. However, it would be good if the paper was sent for proof-read.
Author Response

(The authors gave the same response as above.)

Reviewer 3 Report
Comments and Suggestions for Authors
Thank you for the opportunity to review this manuscript. I would like to leave my questions and suggestions.
Abstract
1-Please, include the date of the study.
2-Lines 20-22: “Eight distinct subject areas have been developed to encompass the data literacy required in healthcare professions. Within these, the assessment of learners' data literacy is structured into four competency areas” Please, describe in the text “Eight distinct subject areas” and “four competency areas”.
3-Please, rewrite the keynotes.
Methods
4-Lines 156-161:” As illustrated in the Figure 1 below, another central component of the methodology involves a workshop with representatives from various disciplines participating in DIM.RUHR (N = 34): Health Sciences (n = 6), General Medicine (n = 5), Health Informatics (n = 4), Software and Systems Engineering (n = 4), Didactics and Educational Research (n = 3), Health Services Research (n = 3), Law (n = 3), Social Sciences (n = 2), Media Education (n = 2) and Information Sciences (n = 2).”
This information could be included in “Results” section.
It seems that this workshop was a focus group with many participants (34). It is confusing.
Is the workshop remote or face-to-face? Please, describe it in the text.
How were the participants invited? Please, describe it in the text.
What about socio-demographic data of the participants? Please, describe it in the text.
5-Figure 1 could be included in “Results” section.
6-Lines 186-189: “In order to ensure that the developed personas precisely reflect the essential skills necessary for effective data handling in different professional contexts, 19 individual interviews and one group interview with members of the targeted health professions (N=21; 71% female) with a mean age of M = 35.95 years were conducted”
Please, include this information in “Results” section.
“19 individual interviews and one group interview…(N=21;…)” This information is confusing. Please, explain it in the text.
Is the interview remote our face-to-face? Please, describe it in the text.
How were the participants invited? Please, describe it in the text.
What about socio-demographic data of the participants? Please, describe it in the text.
7-Please, include the date of each step (figure 2) in the text.
8-How did the authors analyze the results from the workshop and interviews? Was a content analysis performed? Please, describe it in the article.
9-Lines193-195: In this context, the following professions were interviewed: Medical assistants (n = 4), general practioners (n = 5), medical students (n = 4), healthcare students (n = 3) and health services researchers (n = 5).
Please, include this information in “results” section.
10-Lines 206-208: “A total of 15 learning objectives catalogs were identified, representing the training and education regulations for all professional groups addressed in the learning objectives matrix.”
Please, include this information in “Results” section.
11-Lines 234-237: “Although the process depicted in Figure 2 appears to follow a chronological sequence, it is better understood as a reciprocal process. While individual stages build on one another, transitions between stages occurred frequently throughout the entire development process.”
Please, modify the current figure 2 that looks a chronological sequence.
RESULTS
12-Lines 241- 252: “3.1. Working Definition of Data Literacy for the Development of LOM The working definition of data literacy used in this study is based on a synthesis of the definitions provided by Ridsdale et al. [7], Grillenberger and Romeike [20], and Schüller [21]. According to these authors, data literacy encompasses the knowledge, skills, and attitudes/values necessary for the effective planning, execution, and improvement of all process steps involved in deriving valuable insights or making decisions from data. This occurs within a collaborative culture under ethical, legal, and social considerations. The working definition of data literacy was integrated into the LOM in such a way that all subsequently formulated learning objectives are based on this definition. Additionally, the workshop outcomes were aligned with existing frameworks and matrices and incorporated into the LOM. The approach establishes the foundation for the further conceptualization of the LOM, as the fundamental structure is based on this working definition and the workshop’s outcomes.”
Please, this information could be included in “Methods” section.
13-Many results from supplementary data could be described in “Results” section. Example: Table: Overview of the integrated subject areas
14-The current Figure 3 in the manuscript is difficulty for reading.
15-I did not find the limitations of the study. Please, write it
16-I did not find “Conclusion” section. Please, include it.
Author Response

(The authors gave the same response as above.)

Round 2
Reviewer 1 Report
Comments and Suggestions for Authors
The comments have been addressed accordingly.
Author Response
Comment 1: The comments have been addressed accordingly.
Response 1: We are pleased that we were able to incorporate all of your suggested comments and changes. Thank you for your valuable feedback and the time you dedicated to reviewing our manuscript.
Reviewer 3 Report
Comments and Suggestions for Authors
Congratulations! Please, find my last comments.
1-Please, include the date of the study in “Abstract” section.
2- Lines 142-143: “The study was carried out from February 2023 to November 2024” / Lines 283-284: “The integration of identified content into the matrix took place between May and June 2024.” Was the study carried out from February 2023 to November or June 2024? Please, think about this and correct the information.
3-Lines 307-309: “In summary, the integration of the individual steps listed above into the overall structure of the LOM, as illustrated in Figure 2, proceeded as follows:…” OR “In summary, the integration of the individual steps listed above into the overall structure of the LOM, as illustrated in Figure 1, proceeded as follows:…” Please, correct the number.
4-Line 321: “While Figure 2 suggests a chronological sequence, the development process was equally…” OR “While Figure 1 suggests a chronological sequence, the development process was equally… Please, correct the number.
5-Line 354: “As illustrated in the Figure 1 below…” OR “As illustrated in the Figure 2 below…” Please, correct the number.
6-Lines 390-391: “…that the framework should encompass eight distinct subject areas with various subcategories, which can be found in Table 1 below: (1) Fundamentals and general…” / Line 398: “Figure 3. Overview of the integrated subject areas of the LOM.” Is it Table 1 or Figure 3? Please, correct the information.
7-Lines 454-455: “LOM from the first subject area “Fundamentals and general concepts” is illustrated in Figure 3 below” OR “LOM from the first subject area “Fundamentals and general concepts” is illustrated in Figure 4 below” Please, correct the number.
Author Response
Thank you very much for taking the time to review this manuscript. Please find the detailed responses below in the attached document.
